# Enhancing Foreground Boundaries
# for Medical Image Segmentation

**Dong Yang**                                                DONGY@NVIDIA.COM
**Holger Roth**                                              HROTH@NVIDIA.COM
**Xiaosong Wang**                                      XIAOSONGW@NVIDIA.COM
**Ziyue Xu**                                                 ZIYUEX@NVIDIA.COM
**Andriy Myronenko**                              AMYRONENKO@NVIDIA.COM
**Daguang Xu**                                          DAGUANGX@NVIDIA.COM
*NVIDIA*

## Abstract

Object segmentation plays an important role in the modern medical image analysis, which benefits clinical study, disease diagnosis, and surgery planning. Given the various modalities of medical images, the automated or semi-automated segmentation approaches have been used to identify and parse organs, bones, tumors, and other regions-of-interest (ROI). However, these contemporary segmentation approaches tend to fail to predict the boundary areas of ROI, because of the fuzzy appearance contrast caused during the imaging procedure. To further improve the segmentation quality of boundary areas, we propose a boundary enhancement loss to enforce additional constraints on optimizing machine learning models. The proposed loss function is light-weighted and easy to implement without any pre- or post-processing. Our experimental results validate that our loss function are better than, or at least comparable to, other state-of-the-art loss functions in terms of segmentation accuracy.

**Keywords:** Deep learning, medical image segmentation, boundary enhancement.

## 1. Introduction

In last decade, the automated or semi-automated segmentation approaches have been widely developed to identify and parse organs, bones, tumors, and other regions-of-interest (ROI). However, most of the segmentation approaches tend to have difficulty in producing high quality prediction at the foreground boundary areas where appearance contrast of medical images is intrinsically fuzzy, which is usually caused by the scanner settings, respiration, or body motions during the image acquisition procedure.

Recently, the neural network based methods have been deployed for the segmentation tasks (Çiçek et al., 2016; Milletari et al., 2016; Liu et al., 2018; Myronenko, 2018), and have achieved the state-of-the-art performance in various datasets with different image modalities. These model architectures follow a U-shape fashion using convolutional encoders and decoders, which takes images as direct input and output segmentation masks. In addition, these models are trained end-to-end using gradient-based optimization, with the objective of minimizing well-established loss functions, such as multi-class weighted cross-entropy, soft Dice loss (Milletari et al., 2016). Although such loss functions are capable of handling the class-imbalance issues that

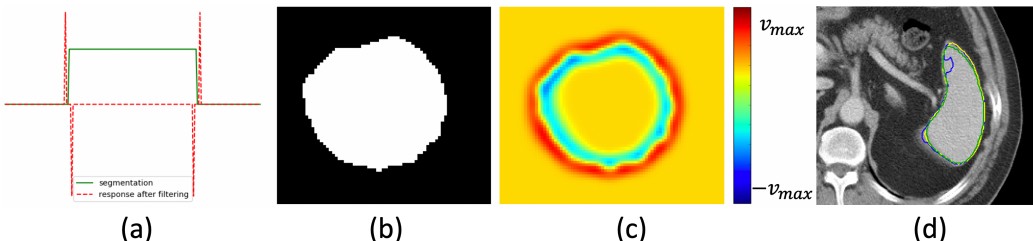

Figure 1: (a). Green curve indicate 1D cross-section of binary mask, and red dashed curve represents the result after filtering; (b). a 2D cross-section of 3D binary mask; (c) a 2D cross-section of 3D output after filtering; (d) Visual comparison of spleen segmentation. Green contour is ground truth label, blue contour is the result applying (Myronenko, 2018), yellow contour is from the proposed work.

often present in the medical image segmentation tasks, the boundary issue is not well addressed, because these functions treat all pixels/voxels equally.

In order to further improve the segmentation performance, we introduce a new loss function, called boundary enhancement loss, to explicitly focus on the boundary areas during training. Our proposed approach shares the similar motivation as to previous work, trying to improve the boundary segmentation of deep neural networks, like (Chen et al., 2016; Oda et al., 2018; Karimi and Salcudean, 2019; Kervadec et al., 2018). Unlike the previous work, our approach is light-weighted without causing much computational burden, and it does not require any pre- or post-processing such as in (Karimi and Salcudean, 2019; Kervadec et al., 2018), or any special network architecture such as in (Chen et al., 2016; Oda et al., 2018) in order to compute the loss function. Furthermore, our proposed loss function is very effective for various segmentation applications, which could be easily implemented and plugged into any 3D backbone networks.

## 2. Methodology

In order to emphasize the boundary regions, we apply the Laplacian filter $\mathcal{L}(\cdot)$, which generates strong responses around the boundary areas and zero response elsewhere, to a 3D binary segmentation mask $S$ in Eq. 1.

$$\mathcal{L}(x,y,z) = \frac{\partial^2 S}{\partial x^2} + \frac{\partial^2 S}{\partial y^2} + \frac{\partial^2 S}{\partial z^2} \tag{1}$$

The discrete Laplacian filtering can be achieved through standard 3D convolution operations. As a result, we can readily compute the difference between filtered output of ground truth labels and filtered output of predictions of a deep neural network. Minimizing the difference between two filtered outputs would implicitly close the gap between ground truth labels and predictions. Following the analysis above, the boundary enhancement loss is defined as a $l_2$-norm shown in Eq.2. Meanwhile, $l_{BE}$ effectively suppresses false positives and remote outliers, which are far away from the boundary regions.

$$l_{BE} = \|\mathcal{L}(\mathcal{F}(X)) - \mathcal{L}(Y)\|_2 = \left\| \frac{\partial^2(\mathcal{F}(X)-Y)}{\partial x^2} + \frac{\partial^2(\mathcal{F}(X)-Y)}{\partial y^2} + \frac{\partial^2(\mathcal{F}(X)-Y)}{\partial z^2} \right\|_2 \tag{2}$$

In practice, the boundary enhancement loss is implemented as a series of single-channel $3 \times 3 \times 3$ convolutional operations without bias terms. Kernels of the first three consecutive convolution layers have identical constant value $1/27$ for smoothing purpose. And the last convolution kernel has fixed values from a standard 3D discrete Laplacian kernel. All parameters of convolution kernels in $l_{BE}$ are non-trainable. The entire operation is similar with the Laplacian of Gaussian (LoG) filtering for edge detection. An example of Laplacian filtering with a ground truth label is shown in Fig. 1.

The overall loss function $l_{overall}$ in our approach is the combination of the soft Dice loss (Milletari et al., 2016) and the boundary enhancement (BE) loss: $l_{overall} = \lambda_1 \cdot l_{dice} + \lambda_2 \cdot l_{BE}$. $\lambda_1$ and $\lambda_2$ are the positive weights between two losses. The boundary enhancement loss cannot be applied alone without the soft Dice loss, because it can not differentiate between the interior and exterior. Take the regions where the label values are constant (0 or 1) for example, everywhere except boundary would be zero after filtering shown in Fig. 1.

## 3. Experiments and Discussion

**Datasets** To cover various objects and image modalities, the datasets of medical decathlon challenge (MSD) (msd, 2018) task 01 (brain tumor MRI segmentation) and task 09 (spleen CT segmentation) are adopted for experiments with our own data split for training/validation. For task 01, 388 multi-channel MRI volumes for training, 96 for validation. And for task 09, 32 CT volumes for training, 9 for validation. Both datasets are re-sampled into the isotropic resolution $1.0\ mm$. For task 01, the voxels are normalized within a uniform normal distribution. For task 09, the voxel intensities of the images are normalized to the range [0,1] according to 5th and 95th percentile of overall foreground intensities.

**Implementation** Our baseline neural network is from (Myronenko, 2018), which has the convolutional encoder-decoder structure using 3D residual blocks. During training, the input of the network are patches with size $224 \times 224 \times 128$ (task 01) and $96 \times 96 \times 96$ (task 09) respectively, randomly cropped from images. $\lambda_1$ and $\lambda_2$ are 1 and 1000 respectively for all experiments. All training jobs use the Adam optimizer. Necessary data augmentation techniques, including random axis flipping and random intensity shift, are used for training. Moreover, the validation follows the scanning-window scheme with small overlaps between neighboring windows. The validation accuracy is measured with the Dice's score after scanning-window inference. The final results are shown in Table 1. Our experimental results show that our proposed approach is capable to work effectively on both structural objects (e.g. organ) and non-structural objects (e.g. tumor). Also, it works well for different modalities of medical images (CT, MRI, etc.). Moreover, our proposed boundary enhancement loss can be easily plugged into any 3D segmentation backbone networks.

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

Table 1: Validation Dice comparison with baseline approaches and proposed approach.

| Method | Task01 | Task09 |
|---|---|---|
| U-Net (Çiçek et al., 2016) | 0.72 | 0.94 |
| AH-Net (Liu et al., 2018) | 0.81 | 0.95 |
| SegResNet (Myronenko, 2018) | 0.83 | 0.95 |
| (Myronenko, 2018)+Boundary Loss (Kervadec et al., 2018) | **0.85** | 0.94 |
| (Myronenko, 2018)+Focal Loss (Zhu et al., 2019) | 0.85 | 0.95 |
| (Myronenko, 2018)+Proposed BE Loss | **0.85** | **0.96** |

Özgün Çiçek, Ahmed Abdulkadir, Soeren S Lienkamp, Thomas Brox, and Olaf Ronneberger. 3d u-net: learning dense volumetric segmentation from sparse annotation. In *International conference on medical image computing and computer-assisted intervention*, pages 424–432. Springer, 2016.

D. Karimi and S. E. Salcudean. Reducing the hausdorff distance in medical image segmentation with convolutional neural networks. *IEEE Transactions on Medical Imaging*, pages 1–1, 2019. doi: 10.1109/TMI.2019.2930068.

Hoel Kervadec, Jihene Bouchtiba, Christian Desrosiers, Éric Granger, Jose Dolz, and Ismail Ben Ayed. Boundary loss for highly unbalanced segmentation. *arXiv preprint arXiv:1812.07032*, 2018.

Siqi Liu, Daguang Xu, S Kevin Zhou, Olivier Pauly, Sasa Grbic, Thomas Mertelmeier, Julia Wicklein, Anna Jerebko, Weidong Cai, and Dorin Comaniciu. 3d anisotropic hybrid network: Transferring convolutional features from 2d images to 3d anisotropic volumes. In *International Conference on Medical Image Computing and Computer-Assisted Intervention*, pages 851–858. Springer, 2018.

Fausto Milletari, Nassir Navab, and Seyed-Ahmad Ahmadi. V-net: Fully convolutional neural networks for volumetric medical image segmentation. In *2016 Fourth International Conference on 3D Vision (3DV)*, pages 565–571. IEEE, 2016.

Andriy Myronenko. 3d mri brain tumor segmentation using autoencoder regularization. In *International MICCAI Brainlesion Workshop*, pages 311–320. Springer, 2018.

Hirohisa Oda, Holger R Roth, Kosuke Chiba, Jure Sokolić, Takayuki Kitasaka, Masahiro Oda, Akinari Hinoki, Hiroo Uchida, Julia A Schnabel, and Kensaku Mori. Besnet: boundary-enhanced segmentation of cells in histopathological images. In *International Conference on Medical Image Computing and Computer-Assisted Intervention*, pages 228–236. Springer, 2018.

Wentao Zhu, Yufang Huang, Liang Zeng, Xuming Chen, Yong Liu, Zhen Qian, Nan Du, Wei Fan, and Xiaohui Xie. Anatomynet: Deep learning for fast and fully automated whole-volume segmentation of head and neck anatomy. *Medical physics*, 46(2):576–589, 2019.

