# OpenReview forum: "Enhancing Foreground Boundaries for Medical Image Segmentation"
_MIDL.io/2020/Conference — MIDL 2020_

### Official Review · AnonReviewer3 · 2020-03-09
**Basically, this is a paper with a simple idea and insufficient experiments.**

**Rating:** 3
**Confidence:** 3

**Review:**

This paper proposes to improve the segmentation quality of boundary areas in medical images. It proposes a loss function that is inspired by Laplacian of Gaussian (LoG) filtering for edge detection. This proposed method is claimed to be light-weighted.

Pros:
1)	This loss function is inspired by Laplacian of Gaussian (LoG) filtering, this formulation is suitable for the task of medical image segmentation.
2)	This paper is well-written.
3)	This loss seems to be easier to implement than competing methods and has competitive results.

Cons:
1)	-- As inspired by LoG, this loss is not novel.
2)	-- The paper says it does not require post-processing. However, the convolution operation seems to be post-processing.
3)	-- The authors argue that this loss is light-weighted, but there is no quantitative evaluation on the computational cost and time consuming compared with other works.
4)	-- The results are only comparable to other methods. It also would be interesting to see the combination of this loss with other methods.

Basically, this is a paper with a simple idea and insufficient experiments. As this is a short paper I recommend weak accept, but it is actually not good enough. I would not be upset if it is rejected.

---

### Official Review · AnonReviewer1 · 2020-03-11
**Elegant loss for boundary refinement**

**Rating:** 4
**Confidence:** 5

**Review:**

This paper is about improving the boundary of a predicted object in the task of image segmentation. Comparing to other losses, they show that they get on par or slightly better performances.

The authors propose a novel loss, that uses the Laplacian filter (which can be implemented efficiencly as successive convolution layers), and then minimize the L2-norm of the filtered output and filtered ground truth. Notice that this doesn't require to define the contour on the continuous softmax predictions, which is often difficult or intractable.

They get very good results on two different datasets.

Many more work and evaluation can be done for that loss, and I am really looking forward to see an more detailed version of this work. But the current form definitely deserve a spot at MIDL2020.

Misc:
- [Kervadec et al. 2018], is actually a MIDL2019 paper

---

### Official Review · AnonReviewer4 · 2020-03-13
**boundary enhancement loss**

**Rating:** 3
**Confidence:** 4

**Review:**

The paper presents a boundary enhancement loss to enforce additional constraints on optimizing machine learning models, which takes the merit of discrete Laplacian filtering and L2 loss to emphasize the boundary regions. The experiment shows the effectiveness of incorporating the proposed loss on brain tumor MRI segmentation and spleen CT segmentation.

---

### Official Review · AnonReviewer2 · 2020-03-13
**The performance cannot convince me that the method works**

**Rating:** 2
**Confidence:** 5

**Review:**

This paper proposed laplacian operator based loss as an extra boundary enhancement loss for the segmentation networks and applied their method on two datasets to make an evaluation.

pros:
1. Since laplacian operator can measure the curvature, laplacian based loss may contribute to boundary enhancement as the paper states.
2. The paper is easy to understand.

cons:
1. Some figures is meaningless, for example, Fig. 1(a).
2. It is reasonable to use series of single-channel 3D convolution to approximate the laplacian operator? Since it is 2nd derivative. Please convince me using either equations or visualizations.
3.The comparison experiments are simple, since there are many boundary-related losses, you only compare with one.
4. The Dice shown in Table I cannot convince me that the laplacian operator is better than the one with boundary loss. Performance is the same in Task 1. For task 09, it seems all networks can achieve high Dice. Then task 09 is hard to prove anything.

Suggestions:
1.Can you please list the standard deviation? Can you please also provide the computational cost?
2. Why UNet's Dice is so low in Task 1?

---

### Meta-Review · Area_Chair1 · 2020-04-06
**MetaReview of Paper278 by AreaChair1**

**Rating:** 3

**Metareview:**

I agree with the majority of reviewers that the use of Laplacian based loss is interesting and the paper is well presented. I recommend the acceptance of this short paper and encourage the authors to integrate the suggestions of the reviewers in their final version.

**Paper Type:**

both

---

### Decision · Program_Chairs · 2020-04-11

Accept